# Iron Homeostasis in the Lungs—A Balance between Health and Disease

**DOI:** 10.3390/ph12010005

**Published:** 2019-01-01

**Authors:** Joana Neves, Thomas Haider, Max Gassmann, Martina U. Muckenthaler

**Affiliations:** 1Department of Pediatric Oncology, Hematology and Immunology, University of Heidelberg, 69120 Heidelberg, Germany; joanamatosneves@gmail.com; 2Translational Lung Research Center Heidelberg (TLRC), German Center for Lung Research (DZL), University of Heidelberg, 69120 Heidelberg, Germany; 3Institute of Veterinary Physiology, Vetsuisse Faculty, and Zurich Center for Integrative Human Physiology (ZIHP), University of Zurich, 8057 Zurich, Switzerland; thomas.j.haider@gmail.com; 4Department of Cardiology, University Heart Center Zurich, University Hospital Zurich, 8091 Zurich, Switzerland; 5Molecular Medicine Partnership Unit, University of Heidelberg, 69120 Heidelberg, Germany

**Keywords:** iron homeostasis, lung diseases, oxygen sensing, hypoxia

## Abstract

A strong mechanistic link between the regulation of iron homeostasis and oxygen sensing is evident in the lung, where both systems must be properly controlled to maintain lung function. Imbalances in pulmonary iron homeostasis are frequently associated with respiratory diseases, such as chronic obstructive pulmonary disease and with lung cancer. However, the underlying mechanisms causing alterations in iron levels and the involvement of iron in the development of lung disorders are incompletely understood. Here, we review current knowledge about the regulation of pulmonary iron homeostasis, its functional importance, and the link between dysregulated iron levels and lung diseases. Gaining greater knowledge on how iron contributes to the pathogenesis of these diseases holds promise for future iron-related therapeutic strategies.

## 1. Introduction

Every cell requires sufficient oxygen to satisfy its demand for oxidative metabolism. Red blood cells (RBCs) are in the center of this process by transporting oxygen from the lungs to every tissue. Efficient oxygen transported by RBCs relies on the presence of hemoglobin, a protein containing four heme groups that have the ability to bind oxygen through their central iron atom. To sustain adequate erythropoiesis, approximately 20–25 mg of iron are required daily [1]. In addition to hemoglobin synthesis, iron is necessary for other fundamental metabolic processes, including DNA synthesis and repair, transcription, and energy production in the mitochondria [2]. Therefore, iron is essential for the biology of almost all organisms. Insufficient intracellular iron levels impair the activity of iron-containing proteins, ultimately compromising cell function and viability. Iron’s critical role is explained by its potential to fluctuate between oxidation states, mainly between divalent ferrous (Fe^2+^) and trivalent ferric (Fe^3+^) iron [3]. However, this chemical property as a transition metal makes free iron very reactive and potentially toxic. Iron catalyzes the production of reactive oxygen species (ROS) via the Fenton and Haber-Weiss reactions [4]. Exposure to these highly reactive radicals damages lipids, nucleic acids, and proteins, causing cell and thus tissue damage. To counteract its chemical reactivity, iron in the body is mainly bound to prosthetic groups, such as heme in hemoglobin or myoglobin, and to proteins, such as the plasma iron transport protein transferrin or the intracellular iron storage protein ferritin. This dual role of iron makes it essential to tightly control iron homeostasis to meet the cellular and systemic metabolic needs while preventing detrimental iron overload. In other words, like oxygen, iron represents a double-edge sword and thus requires exact regulation.

Adequate iron supply and homeostasis depends on systemic plasma availability. Under balanced conditions, only 1–2 mg of iron from dietary sources are absorbed daily by duodenal enterocytes to compensate for small iron losses (e.g., blood loss and desquamation of epithelial surfaces) or increased demand (e.g., pregnancy or hypoxic exposure). Most of the iron required to sustain the body’s metabolic needs is provided by reticuloendothelial macrophages that recycle iron from senescent RBCs [5]. Hepatocytes in the liver are specialized for iron storage, acting as an iron buffer system. Since nature has not come up with a regulated way to eliminate iron from the organism, the coordination of iron fluxes from tissues to circulation is essential to maintain plasma iron levels within a physiological range (10–30 μM) [6]. Plasma iron levels are mainly controlled by the hepcidin/ferroportin regulatory system (Figure 1) [1]. The hepatic hormone hepcidin is secreted in conditions of high systemic iron levels and binds to the only known cellular iron exporter ferroportin (FPN), thereby inducing its ubiquitination, endocytosis, and lysosomal degradation [7,8]. This causes a decrease in iron export from FPN expressing cell-types, such as duodenal enterocytes and iron-recycling macrophages, ultimately leading to a decrease in plasma iron levels.

The lung is the major organ for gas exchange and like all other organs it is exposed to circulating iron. Additionally, lung cells are directly exposed to exogenous iron sources via inhalation (e.g., cigarette smoke). A disruption of iron homeostasis in the lung is frequently associated with respiratory diseases, such as chronic obstructive pulmonary disease (COPD), acute respiratory distress syndrome, and cystic fibrosis [9,10,11]. This review focuses on the regulation of pulmonary iron levels and its functional importance as well as potential intrinsic and extrinsic factors modulating lung iron homeostasis. We will further address the impact of iron in the pathogenesis of human lung diseases as well as other diseases with a pathological lung phenotype associated with alterations in iron metabolism.

## 2. Lung Iron Homeostasis

### 2.1. Balancing Lung Iron Homeostasis

Lungs are composed of numerous cell types that work together to ensure efficient gas exchange between the atmosphere and the bloodstream. As every other cell, lung cells must acquire enough iron to sustain metabolic demand, thus ensuring their survival and lung function. At the same time, lung cells must prevent iron overload, oxidative stress, and resulting injury, which would ultimately impair lung function. The risk of oxidative stress in the lung is increased due to the exposure to physiologically high oxygen levels in the atmosphere. Lung cells are not only exposed to circulating iron but also to iron present in inhaled particles, such as mineral aerosols generated by wind erosion of soils or iron-rich air pollution particles [12,13]. In response to these challenges, the lung maintains protective mechanisms to avoid oxidative damage. The epithelial surface of the respiratory tract is covered by a thin layer of fluid containing high levels of antioxidant molecules, such as ascorbic acid, reduced glutathione, and mucin [14]. Furthermore, human airway secretions contain transferrin and lactoferrin, and glycoproteins are able to bind iron and maintain a chemically inert form [15,16]. Iron bound to transferrin or lactoferrin can be taken up by epithelial cells through their receptors, namely transferrin receptor 1 (TfR1) and lactoferrin receptor (LfR), respectively, to be stored safely bound to ferritin (Figure 2), a multimeric iron storage protein consisting of 24 subunits of heavy and light chains capable of accommodating up to 4000 iron atoms [17,18,19]. Excess of pulmonary iron caused by endogenous or exogenous factors can override these protective mechanisms and cause pulmonary oxidative stress. Experimentally, this was tested by intravenously injecting iron-containing compounds (e.g., iron dextran and ferric carboxymaltose) in rats that subsequently showed high levels of pulmonary oxidative stress, indicated by increased levels of nitrotyrosine and protein carbonyl modifications [20]. Likewise, pulmonary administration of Fe_2_O_3_ nanoparticles via inhalation also generates ROS in the rat lung [21].

### 2.2. Pulmonary Iron Modulation is a Potent Intrinsic Defense Strategy Against Respiratory Pathogens

Respiration not only provides oxygen to support life but also exposes the respiratory system to external pathogens. Iron homeostasis and immune responses are tightly connected, likely reflecting the essential role of iron in the survival, proliferation, and virulence of most pathogens [22]. A central host defense strategy is to limit iron availability for extracellular pathogens by intracellular sequestration [23]. Inflammatory cytokines (e.g., IL6) induce the expression of the iron-regulated hormone hepcidin that degrades the iron exporter FPN located in macrophages and duodenal enterocytes, thereby reducing iron export to the circulation [24]. Additionally, transcription of FPN in macrophages is reduced via the TLR4- or TLR2/TLR6-signalling pathways, further contributing to inflammation-associated hypoferremia [25,26,27]. Indeed, failure to decrease systemic iron levels upon infection strongly increases the susceptibility of hepcidin knock-out (KO) mice to some pathogens, such as Vibrio vulnificus or Klebsiella pneumonia [28]. In line with this observation, individuals with iron overload disorders, such as hereditary hemochromatosis, display a higher susceptibility to various infectious diseases [29,30]. Moreover, supplementation of iron and folic acid in preschool children increased the incidence of malaria and other infectious diseases in Pemba (Tanzania), an area with high infectious burdens [31].

As iron plays a critical role in the development of infectious diseases, the maintenance of low iron levels in the lung is crucial not only to prevent oxidative stress but also to sustain the pulmonary defense against inhaled pathogens. A positive correlation between sputum iron content and *P. aeruginosa* levels in clinically stable cystic fibrosis patients supports this idea [10]. Iron is a strong regulator of *P. aeruginosa* survival and behavior, as 6% of its transcribed genes respond to iron [32]. The success of this Gram-negative bacterium in colonizing the airways is in part due to its ability to form biofilms. Interestingly, several studies showed that iron is necessary for *P. aeruginosa* biofilm formation and application of iron chelators impairs this process by sequestering free iron [33,34]. Based on these findings, pulmonary administration of high-affinity iron chelators via inhalation might emerge as a possible therapeutic approach to fight *P. aeruginosa* lung infections in cystic fibrosis patients [32,34]. Consistently, the incidence of respiratory infections in children with mild-to-moderate iron-deficiency was substantially lower compared to iron-depleted children in Kilimanjaro (Tanzania) [35]. Finally, an association between increased dietary iron intake and increased odds of developing active pulmonary tuberculosis was observed in individuals from Zimbabwe [36].

Patients who underwent lung transplantation showed increased pulmonary iron levels in the allografts after transplantation that possibly contributed to the risk of oxidative stress and lung injury [37,38]. In addition, high iron levels in a tracheal allograft mouse model increased the risk for *A. fumigatus* invasion, a well-known pathogen causing common respiratory infectious disease in lung transplant recipients [39].

During evolution, microorganisms developed high-affinity iron uptake systems, such as siderophores, to acquire iron from the host [22]. The host fights back by increasing the expression of lipocalin-2, a protein mainly produced by neutrophils that binds to the siderophore enterobactin and prevents its uptake by the pathogen [22]. During infection, lipocalin-2 is not only secreted by recruited neutrophils but also from lung epithelial cells [40]. The importance of lipocalin-2 in sequestering iron in the lung is highlighted by the observation that pneumonia caused by intratracheal instillation of *E. coli* is aggravated in lipocalin-2 knock-out mice [40]. Furthermore, lipocalin-2 binding is specific and does not prevent iron uptake and consequent colonization by bacteria that produce modified forms of enterobactin or other types of siderophores [41].

### 2.3. Molecular Regulation of Lung Iron Homeostasis

Iron uptake, utilization, storage, and export must be coordinated to maintain cellular iron homeostasis in every organ. The iron-responsive element (IRE)/iron-regulatory protein (IRP) system plays a central role in this process by controlling the expression of iron-related proteins in response to intracellular iron levels (Figure 3) [42,43]. Iron regulatory protein 1 and 2 (IRP1 and IRP2) interact with conserved hairpin structures named iron-responsive elements (IREs) present in the 5’ or 3’ untranslated regions (UTRs) of mRNAs of iron-regulated genes (Figure 3). In iron-deficient cells, IRPs bind to the IRE in the 5’ UTR of ferritin light chain (FtL), ferritin heavy chain (FtH), FPN, and the transcription factor HIF-2α (see below) mRNAs, inhibiting their translation [44,45,46,47,48]. Additionally, IRPs bind to IREs located in the 3’ UTR of TfR1 or Dmt1 (see below) mRNAs, blocking their degradation [49,50,51]. Subsequently, in conditions of cellular iron deficiency, iron uptake increases while iron storage and export decrease, resulting in higher intracellular iron availability. On the other hand, in iron-loaded cells, IRPs cannot bind to IREs. IRP1 is converted from its RNA-binding form to a cytoplasmatic aconitase containing a 4Fe–4S cluster and IRP2 is targeted for proteasomal degradation [52,53,54]. As a result, iron storage and export are increased and iron uptake decreased.

#### 2.3.1. Control of Pulmonary Iron Uptake

Similar to most cells, lung cells express TfR1 and likely acquire transferrin-bound iron from pulmonary vessels (Figure 2). Increased pulmonary iron levels in mouse models of iron overload are associated with decreased pulmonary TfR1 mRNA levels, suggesting that the IRE/IRP regulatory system controls the expression of iron-related genes in the lung [55,56]. In healthy humans, approximately 30% of plasma transferrin is saturated with iron. In conditions of systemic iron overload, which cause transferrin saturation to rise above 60%, non-transferrin-bound iron (NTBI) accumulates. Exposure to high levels of systemic iron causes accumulation in different pulmonary cell types, such as alveolar macrophages, epithelial cells, and vascular smooth muscle cells [55,56]. So far, the exact iron uptake mechanisms are not understood. The lung expresses the NTBI importers DMT1, ZIP14, and ZIP8, that may further contribute to the iron loading under high systemic and/or local iron levels.

The divalent metal transporter 1 (DMT1) is localized at the brush-border membrane of duodenal enterocytes and was first discovered due to its crucial role in dietary iron absorption (Figure 1) [57]. Dietary inorganic ferric iron (Fe^3+^) needs to be reduced to ferrous iron (Fe^2+^) by the ferrireductase duodenal cytochrome B (DCYTB) before being transported from the duodenal lumen into the cytosol of enterocytes via DMT1 [58]. Airway epithelial cells express both DMT1 and DCYTB, and therefore are likely able to take up NTBI from the airway space (Figure 2) [59,60]. How lung DMT1 expression is regulated in response to pulmonary iron levels is unclear. Different Dmt1 mRNA isoforms are expressed in the lung, including one isoform harboring an IRE in the 3’UTR and another isoform lacking the IRE [61,62]. Increased iron levels inactivate the IRE/IRP system and cause a decrease in the mRNA levels of the IRE-containing DMT1 isoform (Figure 3). Consistently, lower mRNA levels of this isoform are observed in the iron-loaded lung of a murine mouse model of hereditary hemochromatosis type 4 [56]. In addition, lower pulmonary DMT1 mRNA and protein levels were also reported in the iron-loaded lung of hepcidin deficient mice [55]. By contrast, DMT1 protein levels remain unchanged upon lung iron loading via intraperitoneal injection of iron-saccharate [63]. Moreover, another group observed an increase in the non-IRE DMT1 isoform in the rat lung, following instillation with ferric ammonium citrate, without differences in the levels of the IRE-containing DMT1 isoform [64]. Future studies are required to further dissect the molecular mechanisms regulating Dmt1 expression in the lung.

NTBI can also be taken up by cells via ZRT/IRT-like protein (ZIP) 14. Genetic inactivation revealed that ZIP14 is a key player for NTBI uptake by hepatocytes under iron overload conditions [65]. Besides hepatocytes, ZIP14 is expressed by cells in other tissues such as the pancreas and the heart [65,66]. In addition, ZIP14 is detected in airway epithelial cells, and protein levels were shown to increase in the iron loaded murine lung [63]. In 2012, another ZIP protein able to transport iron, named ZIP8, was reported. It is abundantly expressed in the human lung [66]. A disruption in both alleles of the mouse ZIP8 gene causes lethality between the gestational day 18.5 and 48 hours postnatal [67]. Interestingly, the newborns present with anemia and reduced iron levels in some tissues, including the lung [67].

Finally, in conditions of intravascular hemolysis, lung cells can also be exposed to hemoglobin and heme-iron circulating in the bloodstream. When released from RBCs, hemoglobin forms a complex with the glycoprotein haptoglobin in the plasma, which is mainly taken up by reticuloendothelial macrophages; heme released from hemoglobin in the blood forms a complex with the glycoprotein hemopexin and is taken up by liver parenchymal cells [68]. In vivo studies have shown that the uptake of hemoglobin-haptoglobin and heme-hemopexin complexes by the lung is relatively low [69,70,71,72]. Nevertheless, alveolar macrophages express the haptoglobin receptor CD163 and the hemopexin receptor CD91, and therefore are likely able to take up these complexes [73,74]. Alveolar macrophages exposed to heme upregulate the expression of HO-1 that is responsible for heme degradation and protection against the toxic effects of this prosthetic group [75,76].

#### 2.3.2. Iron Storage in the Lung

Lung cells store iron intracellularly bound to ferritin, keeping it in a non-toxic ferric form and thus preventing oxidative damage. Consistent with the role of the IRE/IRP system in controlling ferritin expression, pulmonary ferritin levels increase upon lung iron loading in murine mouse models [55,56]. Higher levels of ferritin were also detected in the bronchoalveolar lavage fluid of individuals instilled with iron-containing particles [77]. When needed, iron stored in ferritin can be made available to the cell. In conditions of low levels of intracellular iron, NCOA4 (Nuclear Receptor Coactivator 4) targets the ferritin complex to degradation in autolysosomes and iron is released into the cytoplasm [78,79]. Genetic inactivation of NCOA4 impairs iron mobilization from stores in several tissues such as the liver and the spleen [80]. Future studies must address the possible role of NCOA4 in the mobilization of ferritin-stored iron in lung cells.

#### 2.3.3. Iron Export from the Lung

The only known iron exporter FPN is expressed in the lung, too, albeit at a much lower level compared to the spleen, the key iron recycling site [55]. Using an in vitro model, it was shown that polarized airway epithelial cells export iron through their apical surface [81]. FPN expression at this site was confirmed using immunohistochemical analysis [55,81]. Additionally, several studies reported an association between increased pulmonary iron content and higher protein levels of FPN in this organ [63,82]. Increased levels of FPN are likely a consequence of the inactivation of the IRE/IRP system and an increase at the transcriptional level. Iron export via FPN is combined with iron oxidation from Fe^2+^ to Fe^3+^, mediated by ceruloplasmin (or hephaestin in the basal membrane of duodenal enterocytes) (Figure 1) [83,84]. Interestingly, ceruloplasmin was detected in human bronchial lavage fluid, suggesting that it might be involved in FPN-mediated iron export in the lung [15]. Note that the pH of lung fluids (e.g., the alveolar lining fluid) may affect the complex process of spontaneous oxidation of ferrous to ferric iron following a sigmoid shaped oxidation rate in relation to pH [85]. Therefore, changes in pH may alter the iron redox-states within the alveolar lining fluid, that in turn would affect trans-epithelial iron uptake and intracellular iron availability.

#### 2.3.4. Pulmonary Hepcidin Expression

Hepcidin is mainly produced and secreted by hepatocytes to control systemic iron levels [1]. Of note, low levels of hepcidin are also detected in other tissues, such as the heart, kidney, and lung [55,86,87,88]. Hepatocyte-specific hepcidin knock-out mice recapitulate the hemochromatosis phenotype observed in constitutive hepcidin deficient mice, indicating that extra-hepatic hepcidin is not enough to maintain systemic iron homeostasis [89]. However, recent studies proposed the existence of cell-type specific regulatory circuitries in some tissues. For example, through the analysis of mice with cardiomyocyte-specific deletion of hepcidin, a cell-autonomous role for cardiac hepcidin in maintaining cardiac iron homeostasis was identified [86]. It is still not clear whether there is a local role of hepcidin in the lung. Some reported that the induction of pulmonary hepcidin expression by bacterial lipopolysaccharide (LPS) instillation does not affect FPN protein levels in the lung [55]. Consistently, others showed that an increase in hepcidin expression in human bronchial epithelial cells in vitro does not cause a decrease in FPN protein levels [90]. By contrast, an increase in lung FPN protein levels was observed upon knocking down hepcidin in airway epithelial cells in a mouse model of sepsis-induced acute lung injury [91]. Overall, future studies are necessary to fully understand the potential role of lung hepcidin in pulmonary iron homeostasis.

#### 2.3.5. Alveolar Macrophages

Alveolar macrophages frequently accumulate iron in lung diseases and conditions of iron overload [11,55,56,63]. While splenic and liver macrophages are essential to recycle iron from aged RBCs, the location of alveolar macrophages in the alveolar space suggests that at least in healthy conditions these cells do not contribute to hemoglobin-derived iron recycling. Instead, alveolar macrophages are expected to have a protective role by scavenging the excess of iron in the lung (derived from inhaled iron particles or from plasma iron) thereby limiting its availability to induce oxidative damage [92]. The expression of TfR, LfR, and DMT1 was reported in alveolar macrophages, indicating that these cells might be able to take up iron-bound to proteins as well as ‘free’ iron from the alveolar space, storing it intracellularly in ferritin (Figure 2) [55,93,94,95]. However, it is still not known which iron-uptake mechanism is predominant under physiological conditions. It is further possible that alveolar macrophages take up iron-rich particles from the alveolar space via phagocytosis. Whether these macrophages express FPN under normal iron conditions is unclear. In earlier work, we did not detect FPN in alveolar macrophages isolated from the bronchoalveolar lavage of wild-type mice by immunocytochemical analysis [56]. Similarly, others reported very low protein levels of FPN in these cells by western blot analysis [55]. In contrast, another group detected FPN at the cell membrane of freshly isolated alveolar macrophages, that was significantly reduced in the presence of hepcidin [93]. Consistent in most studies is the observation that higher levels of FPN were detected in alveolar macrophages in murine and rat models under iron loading conditions [55,56,82]. Finally, alveolar macrophages express hepcidin, which increases upon an inflammatory stimulus, but not upon iron loading [55,93].

### 2.4. Modulating Factors of Lung Iron Homeostasis

Pulmonary iron levels depend on plasma iron availability. Consequently, alterations in systemic iron levels have an impact on lung iron content. Mice injected intraperitoneally with iron-saccharate or rats maintained on a high iron diet increased lung iron levels compared to control animals [63,96]. Similarly, mice with genetic iron overload due to either hepcidin deficiency or a mutation in FPN that confers resistance to hepcidin binding show pulmonary iron accumulation [55,56]. Interestingly, high lung iron content was also observed in liver-specific hepcidin deficient mice [55]. This finding strengthens the hypothesis that lung iron loading occurs as a consequence of increased plasma iron levels and not due to the absence of a functional hepcidin/ferroportin system in the lung. By contrast, an increase in lung iron content was not observed in HFE deficient mice with a less severe form of hemochromatosis [87]. Note that HFE is a positive regulator of hepcidin and in its absence hepcidin expression is attenuated [97].

Patients with thalassemia major are hallmarked with ineffective erythropoiesis and therefore require frequent blood transfusions to ensure an adequate oxygen supply to the body [98]. Red blood cells contain high amounts of iron (1 mg of iron per milliliter of erythrocytes [6]). As a consequence of multiple transfusions and due to the absence of a regulated way to excrete iron, these patients develop massive systemic iron overload [99]. Iron accumulation was detected in alveolar macrophages from these patients further supporting the hypothesis that increased systemic iron levels elevates pulmonary iron content [100,101].

On the other end of the spectrum, iron deficiency does not seem to alter pulmonary iron levels. Iron deficiency induced by repeated bleeding of mice or a low iron diet (20–25 ppm Fe versus control diet with 200 ppm Fe) in rats caused decreased iron content in liver and spleen but not in the lung [63,102]. Contrasting these studies, rats fed for three weeks on an iron-deficient diet (5,9 ppm Fe) showed a reduction in lung iron levels compared with control animals fed on a diet containing 128 ppm Fe [103]. Differences may reflect the degree of systemic iron deficiency achieved in different studies, whereby the lung iron content may only be altered under conditions of severe iron deficiency. It would be of interest to analyze pulmonary iron levels in mice with genetic iron deficiency, such as Tmprss6 knock-out mice, an animal model with severely impaired iron uptake due to elevated hepcidin levels [104]. Note that TMPRSS6 is a well-known negative regulator of hepcidin [105].

Approximately 60 to 70% of the body’s iron is localized in RBCs. The lung vasculature accommodates the entire cardiac output, allowing almost all RBCs to pass through the alveolar capillaries in order to be loaded with oxygen. Conditions that affect alveolar capillary integrity/permeability (e.g., Goodpasture’s disease) [106] can lead to hemorrhage into the alveolar space. These patients show alterations in lung iron homeostasis with increased iron levels in alveolar macrophages, likely resulting from hemoglobin-derived iron from phagocytosed RBCs [107]. Such cases are examples of disorders that modulate lung iron homeostasis but are not a direct consequence of alterations in iron regulatory mechanisms.

## 3. Environmental Factors that Impact Lung Iron Homeostasis

Environmental factors also affect systemic and lung iron homeostasis. For example, global air pollution and exposure to cigarette smoke are well known to alter pulmonary iron homeostasis. Another potent environmental factor is the exposure to high altitude, which is further exaggerated when combined with physical activity, e.g., during mountaineering at high altitude or altitude training. Pollutants from soil and drinking water (e.g., heavy metals) can significantly disrupt iron homeostasis as well. Finally, thermal stress induced by extremely warm or cold environmental conditions may also affect iron metabolism but are not within the scope of the present review.

### 3.1. The Impact of Air Pollution and Cigarette Smoke on Iron Homeostasis

According to a recently released WHO news report [108], it is estimated that 9 out of 10 people worldwide breathe polluted air, resulting in 7 million additional deaths each year. Fine particles from polluted air penetrate the airways and subsequently reach the cardiopulmonary system causing a variety of diseases such as stroke, heart disease, COPD, lung cancer, and pneumonia [108]. These airborne fine particles, including iron-containing and iron-binding particles, accumulate in the lower respiratory tract, e.g., within the bronchial epithelium. Several groups reported that instillation of iron-containing particles in mice and rats induces increased cellular oxidative stress and lung inflammation [21,109].

Apart from air pollution, cigarette smoke also causes fine (iron) particle exposure of the airways and modulates lung iron homeostasis [110]. The cigarette smoke-induced lung injury was shown to be particle related and is clinically associated with a loss of lung function, increased bronchial hyper-responsiveness, and a pathological lung tissue transformation (e.g., emphysema, fibrosis) [111]. The tracheal lavage from rats exposed to cigarette smoke contained higher levels of iron, transferrin and ferritin. Consistently, the lung non-heme iron content was increased. By contrast, serum iron and transferrin concentrations were reduced in these exposed rats. Additionally, serum ferritin and liver non-heme iron concentrations were increased, indicating a combined disturbance of lung and systemic iron homeostasis [110]. A similar expression pattern in the bronchial lavage of smokers with or without COPD was confirmed by the same authors.

Overall, the majority of cigarette smoke induced lung diseases are of inflammatory nature and strongly depend on activated immune-modulatory signaling pathways [112]. It was suggested that particle-induced ROS-driven mitochondrial dysfunction could have a key role in the pathophysiology of air pollution/cigarette smoke related lung diseases [113]. Although the underlying molecular mechanisms of how environmental fine particle exposure affects pulmonary cellular iron homeostasis are not fully understood, there is a fundamental link to the most globally frequent and deadliest respiratory diseases such as pneumonia, COPD, and lung cancer.

### 3.2. Does Environmental Pollution Alter Iron Homeostasis?

The environmental iron content depends on the soil iron content in given geographical regions and often influences iron levels in the drinking water. For example, the Tibetan Plateau contains a higher percentage of soil iron in comparison to the Chinese region [114]. As Tibetans often live from farming and mining, this living habit suggests a higher exogenous exposure to iron. Unfortunately, reports are not available as to whether this affects iron homeostasis and health in Tibetans. On the other hand, it is well-known that other environmental pollutants derived from soil and drinking water, such as persistent organic pollutants, heavy metals, and pesticides, significantly affect body iron homeostasis with toxic effects on organic functions [115]. Finally, the habit of open-fire cooking has to be considered as a source of iron pollution, too. As systemic iron availability directly affects lung iron availability these environmental factors may also impact on lung function and health in exposed individuals. Future research is needed to prove this assumption.

### 3.3. High-Altitude Exposure and Physical Exercise

More than 140 million people permanently live at altitudes of 2500 m above sea level and millions of other people travel to high altitude every year [116]. Adequate iron homeostasis is essential for oxygen uptake, transport, and delivery in the body, ultimately allowing aerobic metabolism. Therefore, it is not surprising that iron metabolism and aerobic metabolism are strongly linked to each other. Humans can, at least to a certain extent, adapt to high altitude and subsequently also to reduced atmospheric oxygen partial pressure levels, as exemplified by the highest permanent human settlements at altitudes of 5500 m above sea level at the Qinghai-Tibetan plateau in Asia [117]. On the other hand, exposure to high altitude can also lead to pathological maladaptations, which result in the well-known acute and chronic forms of high-altitude-related diseases [118].

At the systemic level, the mechanisms by which the human body can adapt to high altitude by altering iron acquisition have been recently discussed. This involves key organs of iron homeostasis such as the liver, the bone marrow, and the small intestine (duodenum), as well as signaling molecules such as erythropoietin, hepcidin, and erythroferrone that interact to control plasma iron levels [119]. Briefly, the recently discovered erythroid regulator erythroferrone (ERFE) [120] is released mainly from erythropoietin (Epo)-stimulated erythroblasts to suppress hepcidin production in hepatocytes resulting in increased iron absorption in the duodenum and iron export from reticuloendothelial macrophages. This elevates systemic iron availability required for RBC synthesis [121]. Higher levels of circulating RBCs improve tissue oxygenation by increasing blood oxygen transport capacity to compensate for the reduced alveolar oxygen diffusion at high altitude.

Oxygen demand increases upon physical exercise at high altitude. As a consequence, the iron turnover increases by a combination of an elevated iron demand due to exercise induced enlargement of red blood cell volume and an enhanced iron loss due to exercise induced sweating, hemolysis, hematuria, gastrointestinal bleeding, and, as recently reported, a modification of hepcidin levels [122]. Furthermore, iron deficiency is quite common in athletes, especially in females, and significantly compromises exercise performance [123], especially when performing at high altitude [124]. A very recent in vitro study found that iron chelator deferoxamine (DFO)-induced intracellular iron deficiency reversibly compromises contractility and relaxation of human cardiomyocytes by a significant impairment of mitochondrial respiration [125]. This indicates a direct impact of iron deficiency on cardiac function.

At the pulmonary level, convincing evidence exists for a crucial role of iron homeostasis in the maintenance of physiological lung function as well. This becomes particularly evident under stress conditions, such as high-altitude exposure. Here, the functional units of the cardiopulmonary system, namely the airways, the pulmonary vasculature, the respiratory control, the respiratory muscles, and the heart must interact in a precise and integrative manner. Balanced iron availability is important for every single functional compartment of the cardiopulmonary system to successfully adapt to high altitude.

Apart from the previously mentioned impact of cellular iron scavenging on human cardiac muscle cell function, other experimental studies showed that the chronic application of the iron chelator ciclopirox olamine (CPX) negatively affected the acute hypoxic ventilatory response of rats [126,127]. This response is crucial for high altitude adaption and is primarily mediated by the carotid bodies, the main arterial oxygen-sensing organs [128]. Of note, structural changes and altered HIF-1α expression (see below) were found in the tissue of harvested carotid bodies of CPX-treated rats [129,130].

The airways and the pulmonary vasculature must also respond in an orchestrated manner to optimize oxygen transfer and gas exchange within the lungs at high altitude and the functional quality of this process is reflected by an improved ventilation/perfusion (V/Q) balance [131]. Physiological studies clearly demonstrated the importance of iron balance for the hypoxic pulmonary vasoconstriction (HPV), also known as the Euler-Liljestrand reflex, and generally for the response of the pulmonary vasculature to acute and sustained hypoxia [132]. By infusing iron or the iron chelator DFO in healthy adults, the authors showed that iron infusion can abolish the HPV sensitization after re-exposure to hypoxia. In contrast, DFO application increased the effect. An exaggerated HPV is associated with high altitude-related diseases (see disease section). Notably, the ventilatory response of the study subjects to acute hypoxia was also affected by the different treatment interventions [132,133].

Finally, the Tibetan highlanders living on the Tibetan plateau represent the most successful human example of high-altitude adaption, having a unique setting of a potentially replenished iron status [134], an increased pulmonary capacity, and a blunted hypoxic ventilatory response [135], a naturally-selected and genetically-based hypo-responsive HIF-signaling system [136], and a superior physical capacity at high altitude, which led to their nomination as ‘King of the mountains’ [137].

## 4. The Role of Heterodimeric Hypoxia-Inducible Factors 1 and 2 (HIF-1 and HIF-2) in the Lung

During mammalian lung development, a close interaction between airway epithelium and vascular endothelium occurs, which is driven by hypoxic stimuli. The HIF-VEGF-axis is essential for this process. HIF-1 and HIF-2 are heterodimeric transcriptional key regulators of epithelial, mesenchymal, and vascular lung morphogenesis, which are composed of an oxygen-regulated α-subunit and a constitutively expressed β-subunit, the latter also named ARNT. HIF-1α expression is known to dominate early lung development, while HIF-2α up-regulation starts at the saccular stage. Although global deficiency of HIF-1α results in lethality at embryonic day 11 (E11) in mice, the deletion of HIF-2α causes defective lung development, decreased surfactant production, postnatal respiratory distress, and neonatal lethality [138,139].

Several recent reviews have discussed the cellular interference between iron- and oxygen-sensing within the lungs [133,135,140,141,142]. Mammalian cells seem to have multiple molecular structures for sensing intracellular iron and oxygen levels of which some also act as metabolic sensors, while the mitochondrion seems to be the central sensing organelle. The process of intracellular oxygen sensing and signaling has been described in great detail recently [143,144,145] and thus these mechanisms will not be repeated here.

Although the prolyl-hydroxylase domain (PHD)-HIF-oxygen-sensing cascade and the IRP-IRE-signaling machinery are ubiquitously expressed within mammalian organs including the lungs [146,147], the cellular interferences between the two pathways as shown in other organs (e.g., liver and kidneys) have not yet been translated to the lungs. However, the responses of the cardiopulmonary system to alterations of iron- and oxygen availability as reported previously, strongly suggest their presence also within the lungs.

First of all, ferrous iron (Fe^2+^) is a critical co-factor apart from the presence of oxygen, ascorbate, and 2-oxoglutarate (2-OGH) for the function of PHDs (PHD1-3), which act as intracellular oxygen sensors that require iron to regulate HIF-degradation [143]. The PHD2-HIF2-signaling pathway is believed to play an important role in the hypoxic response of the lungs [146] and is affected by alterations in iron availability. In low oxygen conditions, IRP1 activity is reversible reduced while IRP2 activity is increased, which suggests a different role of the two iron regulatory proteins (IRPs) in this context [140]. It was shown that during hypoxic exposure HIF-1α can interact with the hypoxia response elements (HREs) of the IRP1 5’-regulatory region and down-regulate IRP1 expression (HIF-1α-IRP1-axis) [148]. The increase in the levels of IRP2 upon hypoxic exposure is caused by inhibition of its degradation mediated by the iron- and oxygen-dependent F-box and Leucine Rich Repeat Protein 5 (FBXL5) [43,149]. As mentioned earlier, HIF-2α mRNA has an IRE in its 5’ UTR, allowing the translational regulation of HIF-2α via IRPs, dependent on intracellular iron availability [48]. The HIF-2α-IRE primarily interacts with IRP1 [150]. The IRP1-HIF-2α-signaling axis is proposed to be one of the most important links between intracellular iron homeostasis and oxygen sensing [151].

We and others [147] speculate that this mechanism also occurs in the lungs and is potentially involved in the response of the pulmonary vasculature and the airways to acute and sustained hypoxia. Early on, it was shown that the exposure of cells to the iron chelator DFO induces HIF-1α expression [152], suggesting that intracellular iron deficiency has an hypoxia-mimicking effect. It is now well-known that several HIF target genes are involved in the regulation of iron homeostasis. For example, HIFs activate the transcription of genes encoding for TfR1, DMT1, heme oxygenase 1 (HO-1), FPN, and ceruloplasmin [144]. Additionally, during hypoxia, HIF-1α induces the microRNA miR-210, which represses the iron sulfur cluster units 1 and 2 (ISCU1/2), proteins that facilitate the assembly of Fe-S clusters [153,154]. Thus, the miR-210-ISCU1/2-Fe-S cluster axis is proposed to be another important interference pathway between intracellular iron homeostasis and oxygen sensing within the lungs.

## 5. Linking Lung-Related Diseases to Disrupted Lung Iron Homeostasis

A growing number of acute and chronic lung diseases as well as other diseases with a pathological manifestation within the lung are associated with disrupted lung iron homeostasis, leading to either iron deficiency or iron overload (summarized in Table 1). These diseases were also reviewed by other research groups, albeit with a different primary focus [19,133,135,155,156,157]. Generally, the lungs are not considered as a primary iron regulating/storing organ, especially in contrast to the liver, the skeletal muscles, the duodenum, the reticuloendothelial system, and the bone marrow including the RBC pool [2]. However, the regulation of iron homeostasis within the mammalian lungs is tightly controlled (iron content range: 0.4-0.9 mg/g lung tissue according to [155]) in order to maintain proper lung function and also to adapt to changes in iron needs upon changing body conditions, e.g., during exposure to environmental stress as experienced at high altitude. Thus, the iron balance within the lungs is crucial for health and disease. Below we mention selected lung and lung-related diseases in which a link to disturbed iron homeostasis was established. It is beyond the scope of the current review to mention all pulmonary diseases in which iron eventually plays a role.

### 5.1. Acute High-Altitude Illnesses and High-Altitude Pulmonary Edema (HAPE)

The classical high-altitude illnesses are also comprised under the term ‘mountain sickness’. Acute mountain sickness (AMS) is a common manifestation of high-altitude illness and related to a cerebral intolerance to hypoxia. The same is the case in high-altitude cerebral edema (HACE), a severe and life threating condition [167,190]. In the lungs, the development of high-altitude pulmonary edema (HAPE) can occur in unacclimatized healthy individuals at high altitude (≥ 2500 m above sea level) within 1–5 days upon arrival [167,190,191]. The early clinical symptoms are excessive exertional dyspnea, mild cough, chest tightness, and reduced exercise capacity, which can continue to worsen with dyspnea at rest and cough as the edema progresses. In the advanced stage, it can lead to gurgling sounds in the chest and pink frothy sputum. The chest radiograph shows a patchy to confluent edema distribution. The bronchoalveolar lavage shows protein-rich exudate and mild alveolar hemorrhage. Although the exact pathophysiology is not fully understood, it is widely accepted that HAPE is primarily induced by hemodynamic changes within the pulmonary circuitry [167,190,191]. A recent field study reported elevated serum hepcidin levels in HAPE patients exposed to high altitude in comparison to control subjects, that was correlated with higher levels of IL6 [168]. Although no correlation between serum iron levels and pulmonary arterial systolic pressure (PASP) was found in that study, the cellular iron availability within the lungs might have been altered in these HAPE patients, which is critical for the hypoxic pulmonary vasoconstriction (HPV) response. Interestingly, the baseline PASP values in the later HAPE patients were already significantly elevated when compared to healthy controls, indicating a different pre-setting of the pulmonary vasculature tone. Of note, C-reactive protein (CRP) levels were also significantly increased in HAPE patients at high altitude, which may have interfered with serum ferritin levels but also other iron parameters and thus these values have to be interpreted with caution [192]. Overall, an a priori iron status check before ascending to high altitude may help mountaineers to optimize the acclimatization process leading to an increased exercise performance and potentially prevent an iron depleted status with eventually negative health consequences at high altitude.

### 5.2. Chronic Mountain Sickness (CMS)

CMS or Monge’s disease is a highly prevalent disease in high-altitude residents such as the Andeans, with exception of Tibetans where CMS is very rarely seen [193]. The hallmark characteristics of CMS are accentuated hypoxemia, excessive erythrocytosis (women: Hb ≥ 190 g/L, men: Hb ≥ 210 g/L), and pulmonary hypertension, which often results in right ventricular enlargement and hypertrophy, and in the later disease stage, leads to chronic heart failure. Therapy usually comprises of regular phlebotomy [163]. Iron deficiency itself, e.g., induced by the iron chelator DFO, is associated with an elevated pulmonary vascular resistance (PVR), as observed during exposure to hypoxia [194]. In this context, two randomized placebo-controlled trials were performed [164]. In the first trial, they showed that a single infusion of iron lead to 40% reduction of the pulmonary hypertensive response in healthy sea-level residents after exposure to high altitude (4340 m above sea level). In the second cross-over trial, they could show that repeated isovolemic venesection (2 L of blood) was associated with a 25% increase in PASP in CMS patients. Although the subsequent iron replacement did not acutely reverse the effect of iron deficiency on PASP in their setting, the authors argued that the chosen time window might have been too short and further concluded that careful adjustment of iron balance might be a promising strategy to ameliorate the severity in CMS. In summary, it still needs to be proven by future clinical trials whether tight iron status monitoring eventually in combination with therapeutic iron correction improves disease severity and quality of life in CMS patients.

### 5.3. Pulmonary Hypertension (PH)

PH is a complex pathophysiological state characterized by a mean pulmonary artery pressure above 25 mmHg at rest, assessed by catheterization of the right heart. PH is currently subdivided into five different groups [195] and can lead to pulmonary vascular remodeling, right ventricular hypertrophy, and right heart failure (for further detailed clinical information, see [179,196]). A central role of an imbalance between iron homeostasis and oxygen sensing in the pathophysiology of PH was recently discussed [180,181]. The authors proposed a model of combined inflammation driven (via the IL-6-STAT3-hepcidin axis) iron deficiency and exaggerated activation of HIFs. In line with this, it was shown that IRP1^-/-^ mice develop polycythemia and PH via translational de-repression of HIF-2α [182]. Moreover, the same authors also reported that the cultured pulmonary endothelial cells of IRP1^-/-^ mice showed elevated HIF-2α expression levels. Additionally, rats fed on an iron deficient diet have higher levels of HIF-1α and HIF-2α in the lungs, which was associated with an increased pulmonary artery pressure (PAP) and pulmonary vascular resistance [197]. In humans, it was reported that iron deficiency is quite common in idiopathic PH patients [183]. Recent association studies between iron deficiency and PH prevalence suggest that the PH-subtype and eventually also ethnicity may play role in this context [198,199]. These studies together, with the previously reported applied lab studies [132] and clinical trials in humans [164,200], clearly suggest a protective effect of iron against PH and hypoxia-related diseases.

On the contrary, strong positive iron shifts, e.g., in iron overload conditions seem to induce opposite effects, potentially via increased oxidative stress [201]. Indeed, a very recent animal study found that chronic iron overload of rats via daily iron-dextran injections (i.p.) for 28 days induced vascular hyperreactivity and inward remodeling of pulmonary arteries, as well as heart dysfunction [202]. Interestingly, it was recently found that miR-210 was up-regulated in several PH animal models and that the inhibition of mir-210 significantly reduced pulmonary pressure and vascular remodeling in a PH animal model [203], further suggesting a significant role of the miR210-ISCU1/2-Fe-S cluster axis in the PH pathophysiology [184]. Although the lungs seem to have a well-balanced ‘buffer capacity’ for iron, an excessive disruption of iron balance in either direction has detrimental consequences on cardiopulmonary function. Overall, iron homeostasis and HIF oxygen-sensing seem to be crucial in the pathology of PH. Accordingly, therapeutic modifications of these interacting signaling pathways hold great promise for PH patients.

### 5.4. Chronic Obstructive Pulmonary Disease (COPD)

COPD is meanwhile the third leading cause of death worldwide, inducing a significant global socioeconomic burden. Tobacco smoking is the most well-known risk factor for the development of COPD and also for lung cancer. COPD is typically marked by chronic respiratory symptoms and airflow limitations, which are caused by small airways disease and parenchymal destruction (emphysema). The key symptoms of COPD are shortness of breath and dyspnea, as well as chronic cough and sputum [165,204]. COPD pathophysiology is complex, however chronic systemic inflammation potentially triggered by cigarette-induced ROS formation is suspected to be a central aspect of it [205]. Considering the main cause (cigarette smoke) and patho-mechanism (chronic inflammation), it is highly suggestive that COPD is also linked to a disturbance of iron homeostasis. The special setting in COPD of local iron containing and non-containing particle load of airways and potential systemic iron deficiency would suggest a shift towards a mixed iron status. Importantly, cigarette smoke not only contains a certain quantity of iron particles but also over 4000 chemicals of which 100 are known carcinogens and 900 suspected carcinogens. A recent large cohort study found that in both smokers and non-smokers, certain markers of iron homeostasis, such as serum ferritin concentration, serum iron concentration, and transferrin saturation, were associated with critical parameters of lung function (e.g., positive correlation for all three markers in both groups combined with forced vital capacity and forced expiratory capacity) [206]. Another indicator for the presence of disrupted iron homeostasis is the high prevalence of anemia and systemic iron deficiency in COPD patients. The prevalence of anemia in two different cohorts of COPD patients ranged between 23% to 33% [207,208]. In another COPD patient cohort, it was found that iron deficiency was associated with an increased frequency of self-reported exacerbations and reduced exercise capacity [209].

It has been recently proposed that in patients with COPD, pro-inflammatory cytokines (e.g., IL-6) trigger increased hepatic hepcidin expression and secretion [155]. This causes FPN degradation and subsequently reduces cellular iron export into the blood stream, likely resulting in the systemic iron deficiency and anemia observed in these patients. On the other hand, alveolar macrophages from COPD patients were shown to accumulate iron and the percentage of iron-loaded macrophages increased with disease severity [11]. Another team demonstrated that IRP2^-/-^ mice were protected from cigarette smoke-induced experimental COPD [166]. These authors also demonstrated that IRP2 increased mitochondrial iron content and concentrations of cytochrome c oxidase (COX), resulting in impaired mitochondrial function and subsequently COPD in cigarette smoke-exposed mice. On the contrary, mitochondrial iron chelator or a low-iron diet protected mice from cigarette-induced COPD. Overall, regular monitoring of iron status and eventual therapeutic modulation of local and systemic iron homeostasis may avoid the negative consequences of lung iron overload and systemic iron deficiency in COPD patients.

### 5.5. Asthma

Asthma and COPD belong to the subgroup of obstructive pulmonary diseases [158]. The potential role of a disrupted iron homeostasis in asthma was reviewed in great detail recently [159]. Briefly, the author reported profound evidence of disturbed iron homeostasis in asthma by systematically evaluating the known demographic (e.g., ethnicity), physiologic (e.g., exercise), and pathologic (e.g., infections) asthma determinants in the context of iron availability and concluded a setting of absolute or functional iron deficiency in asthma, which may have therapeutic implications in the future.

### 5.6. Cystic Fibrosis (CF)

CF is a severe genetic pulmonary disorder triggered by impaired function of the anion transporter CF transmembrane conductance regulator (CFTR) that results in increased secretion of abnormally viscous mucus and at the clinical level, CF is characterized by chronic bacterial airway infection (e.g., *Pseudomonas aeruginosa*), prominent neutrophilic inflammation, mucus accumulation within the airways, and progressive bronchiectasis (formation of irreversible airway enlargements). The structural changes within the lungs can be visible early using chest imaging. The host-defense defect in the airways of CF patients plays a central role in the disease pathophysiology and leads to chronic airway infections with inflammatory driven airway remodeling [161]. Evidence of disturbed iron homeostasis in CF was found in experimental CF models and also in humans suffering from CF; a setting of systemic iron deficiency and abnormal local iron sequestration within the airway cells were reported [156]. Indeed, elevated iron levels were found in the bronchoalveolar lavage, sputum, macrophages, and in explanted lung tissue from CF patients compared to healthy controls [162]. Furthermore, an increased expression of ferritin, DMT1, and FPN was found in the lung tissue of these patients. Overall, the reported disruption of host iron homeostasis within the lungs of CF patients might be a therapeutic target, especially to reduce chronic airway infections [210]. Alternatively, a recent study showed that it is also possible to directly target bacterial iron homeostasis to control airway *P. aeruginosa* infections via the administration of the metal gallium [211]. Gallium is taken up by bacteria instead of iron because bacterial uptake systems cannot differentiate between them. Gallium is then incorporated into iron-containing proteins, but, since it cannot be reduced under physiological conditions, it impairs their normal activity and therefore bacterial survival and proliferation.

### 5.7. Lung Cancer

As iron is an indispensable component for cell metabolism, proliferation, and growth, it is equally crucial for tumor metabolism, tumor proliferation, and tumor growth [171]. In fact, population-based studies suggest that higher levels of overall body iron are linked to a higher risk of developing cancer [172]. In particular, subjects with transferrin saturation level above 60% were reported to have increased risk to develop lung cancer [212]. It was proposed that cancer cells reprogram their cellular iron metabolism towards an increase in cellular iron uptake and decrease of iron export [172]. Briefly, cancer cells increase iron uptake by increasing the expression of TfR1 and also via increased secretion of lipocalin-2. Tumour cells expressing TfR1 were detected in 88% of patients with non-small-cell lung cancer (NSCLC), while TfR1 was not detected in the tumor stroma [173]. Furthermore, it was shown that the microRNA miR-20a is inversely correlated with FPN expression in NSCLC cells and that low cellular FPN expression stimulates tumor cell proliferation and colony formation, potentially via increased cellular iron availability [174]. In parallel, a reduction in FPN protein levels in tumor cells might also result from increased hepcidin expression in these cells, probably acting in an autocrine manner [172]. Together, these modifications of cellular iron homeostasis lead to an increased labile iron pool (LIP) within cancer cells.

The role of the IRE/IRP system in regulating intracellular iron levels in tumor cells is still incompletely understood [172]. However, it is worth mentioning that IRP2 was suggested to have pro-oncogenic activity in lung cancers based on tumor tissue microarray analyses [175]. Lung adenocarcinomas, as numerous other cancers, induce tumor-associated inflammation, a mechanism that is triggered by activating central inflammatory pathways, such as the NFκB-signaling pathway. NFκB signaling is important for tumor cell proliferation, apoptosis, metabolism, as well as for tumor angiogenesis and metastasis. NFκB itself can induce the HIF-signaling pathway in both, normoxic and hypoxic conditions. Conversely, HIFs can modulate NFκB-signaling in a bidirectional manner [213].

Overall, the strong iron dependency of lung cancer offers the opportunity for treatment interventions that target iron availability (e.g., iron chelators); however, future pre-clinical and clinical studies are needed to address this potential treatment. In a different perspective, iron modulation in tumor-associated macrophages (TAMs) might also emerge as a possible therapeutic target. Recent studies indicate that the iron loading of TAMs can differentiate them towards a pro-inflammatory phenotype and inhibit tumour growth [176,214]. Supporting this idea, the presence of iron loaded TAMs correlates with reduced tumor size in patients with non-small cell lung cancer [176].

### 5.8. Other Diseases

Acute respiratory distress syndrome (ARDS) is clinically characterized by severe hypoxemia resulting from pulmonary gas exchange failure [160]. A potential source of iron in ARDS may derive from ‘low grade bleeding’ and hemorrhage, which might be present in certain subsets of this heterogenous pathological lung condition [215]. Analysis of the bronchoalveolar lavage of patients suffering from ARDS revealed the presence of higher levels of iron and iron-related proteins, such as ferritin, lactoferrin, and transferrin, likely reflect disturbances in lung iron homeostasis [9]. Similarly, patients with pulmonary alveolar proteinosis (PAP), a disease defined by severe accumulation of surfactant in the airspaces and hypoxemia [177], also present higher concentrations of iron, ferritin, transferrin, and lactoferrin in the bronchoalveolar lavage [178]. Finally, patients with idiopathic pulmonary fibrosis (IPF), a restrictive lung disease with a prominent gas diffusion limitation [169], have a higher frequency of HFE allelic variants associated with the iron overload disease hemochromatosis, when compared to healthy controls [170]. Hemosiderin accumulation in alveolar macrophages was increased in these patients. Whether these alterations in pulmonary iron homeostasis contribute to the pathology of these diseases is still unknown.

Pulmonary dysfunction, in particular restrictive lung disease, is frequently observed in patients with thalassemia major (TM), a disease characterized by severe iron accumulation [186,187,188,189]. It has been speculated that abnormal lung function in those patients might be a consequence of chronic iron overload [189]. Supporting this hypothesis, a restrictive pattern was observed in a mouse model of iron overload, caused by a mutation in FPN that confers resistance to hepcidin binding [56]. In addition to thalassemia, other hemolytic diseases, such as sickle cell disease, are also associated with increased susceptibility to pulmonary diseases and abnormal lung function [216,217,218]. However, future studies are needed to fully understand the impact of iron overload in lung function in humans and whether additional factors such as hemoglobin and heme release in these hemoglobinopathies contribute for the lung disorders observed in these patients.

## 6. Therapeutic Potential of Iron Modulation

The profound numbers of lung diseases associated with disrupted iron homeostasis offer a great potential for the therapeutic careful use of iron modulators. Before corrective interventions can be applied in patients, we require a better knowledge of iron’s role in pulmonary disease and markers to properly evaluate the systemic and pulmonary iron status. Currently, dietary interventions, phlebotomy, blood and iron infusions, as well as iron chelator treatment, are used to correct systemic disturbances of iron homeostasis. For example, blood infusions and iron-chelation therapy are standard in the treatment of hemolytic anemias. In PH patients, the therapeutic correction of the frequently observed iron deficiency may ameliorate the disease outcome and clinical trials are ongoing to further investigate this potential [180]. The first trial results seem to support this concept, however so far only for the idiopathic PH-subtype [185].

Future approaches will directly target regulatory molecules of iron metabolism by applying small agonistic/antagonistic molecules. It is tempting to speculate that such molecules might be applied by inhalation. For example, hepcidin as the major regulator of iron metabolism and its receptor FPN are promising targets for future drug development [219]. Indeed, hepcidin agonists are currently under development to treat iron overload diseases such as hereditary hemochromatosis. On the other hand, hepcidin antagonists are developed to treat inflammation-induced anemias.

Moreover, diseases such as COPD that are characterized by systemic iron deficiency and lung iron accumulation would benefit from a more specific strategy. For example, inhalation of chelators (or iron sources, in case of lung iron deficiency) would have the potential to directly modulate pulmonary iron homeostasis while hopefully having no or only a mild effect on systemic iron homeostasis. Apart from direct targeting of regulators of iron metabolism, indirect targeting of signaling pathways, such as the HIFs, may be an option [220,221]. In summary, therapeutic possibilities to control iron metabolism are widespread, and the pros and cons of these novel therapeutic approaches need to be carefully evaluated.

## 7. Conclusion and Outlook

In healthy conditions, pulmonary iron homeostasis is tightly controlled to maintain proper lung function. In environmental (e.g., high-altitude exposure) and behavioral (e.g., exercise) stress conditions it requires adaptation. Importantly, numerous acute and chronic respiratory diseases are associated with disrupted iron homeostasis in the lungs. The close cellular interaction between iron regulatory pathways via IREs/IRPs or hepcidin/ferroportin and the oxygen sensing pathway via HIFs seems to be critical for healthy adaption but also for pathologic maladaptation within the lungs. Enormous progress has been made in our molecular understanding of these pathways and their suppression and/or enhancement. However, this knowledge currently needs to be applied to the lung. An improved understanding of iron trafficking and storage in the lung and its role in lung disease onset and progression will improve interventional modification of iron homeostasis within the lungs via iron-modulators.

## Figures and Tables

**Figure 1 pharmaceuticals-12-00005-f001:**
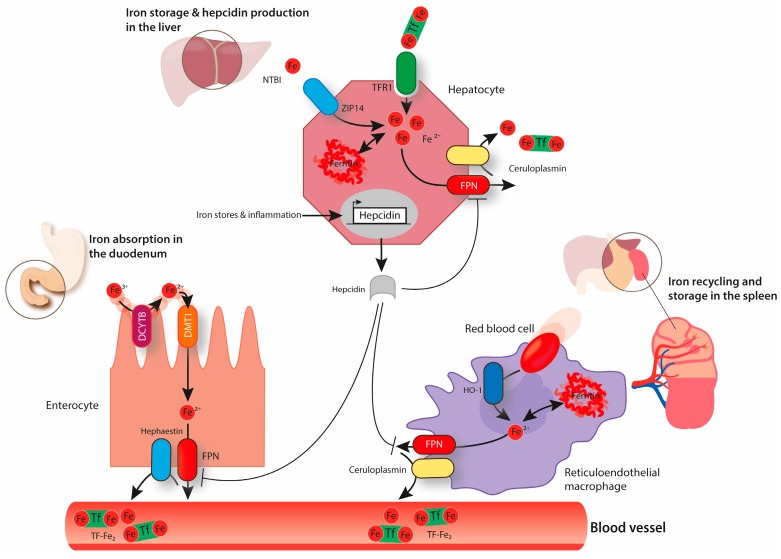
The above image represents systemic iron homeostasis. Dietary iron absorption occurs at the brush-border membrane of duodenal enterocytes. Ferric iron (Fe^3+^) is reduced to ferrous iron (Fe2^+^) by duodenal cytochrome B (DCYTB) and is transported across the membrane via divalent metal transporter 1 (DMT1). The export of iron through the basolateral membrane of enterocytes occurs via ferroportin (FPN) and is coupled to the reoxidation of Fe^2+^ to Fe^3+^, a process that is catalyzed by hephaestin. Ferric iron circulates in the blood bound to transferrin (Tf-Fe_2_). Transferrin-bound iron can be taken up via transferrin receptor 1 (TfR1) by every cell type in the organism, including hepatocytes that store high amounts of iron in ferritin. Hepatocytes can also take up non-transferrin-bound iron (NTBI) via ZRT/IRT-like protein (ZIP14). When required, iron can be exported from hepatocytes via FPN back to circulation (a process combined with the re-oxidation of Fe^2+^ to Fe^3+^ mediated by ceruloplasmin). Senescent erythrocytes are engulfed by reticuloendothelial macrophages. Iron is released from heme by heme oxygenase (HO1) and it can be either stored in ferritin or exported back to the circulation, depending on systemic iron requirements. Iron export from macrophages via FPN is also coupled to the activity of ceruloplasmin. Hepcidin produced by hepatocytes has the ability to decrease cellular iron export by binding to FPN and inducing its endocytosis and degradation. The expression of hepcidin is controlled by several factors including body iron stores and inflammation.

**Figure 2 pharmaceuticals-12-00005-f002:**
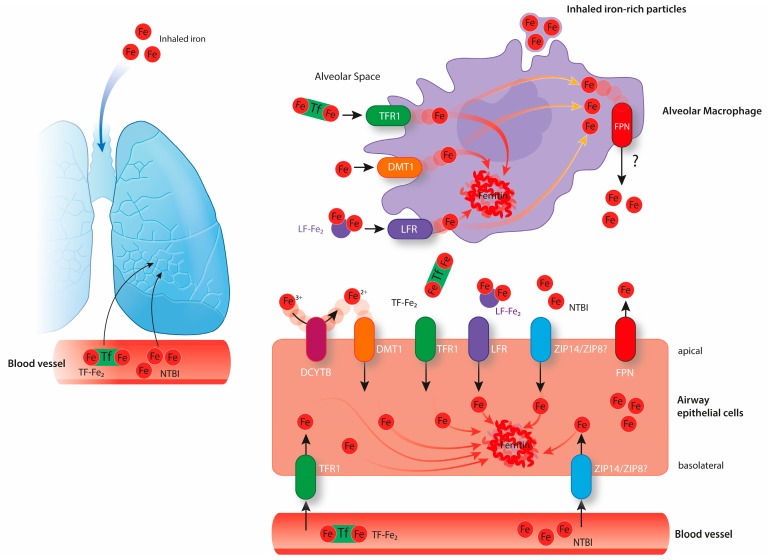
The above image shows pulmonary iron homeostasis. Lung cells are exposed to iron circulating in the bloodstream and to exogenous iron sources via inhalation. Epithelial cells likely take up the iron required for their metabolic needs from the lung vasculature via TfR1. Additionally, airway epithelial cells also take up iron from the airway space via TfR1, LFR, DMT1 (associated with the reduction of Fe^3+^ to Fe^2+^ by DCYTB), and possibly via ZIP14 or ZIP8. These cells store iron intracellularly bound to ferritin or export it via FPN expressed at their apical surface. Alveolar macrophages may take up free iron as well as iron bound to proteins from the alveolar space via DMT1, TFR1 or lactoferrin receptor (LFR). They might further take up inhaled iron-rich particles via phagocytosis. Alveolar macrophages are believed to be crucial in the maintenance of lung iron homeostasis by storing high amounts of iron intracellularly bound to ferritin. It is still not clear if alveolar macrophages express FPN and export iron under physiological conditions.

**Figure 3 pharmaceuticals-12-00005-f003:**
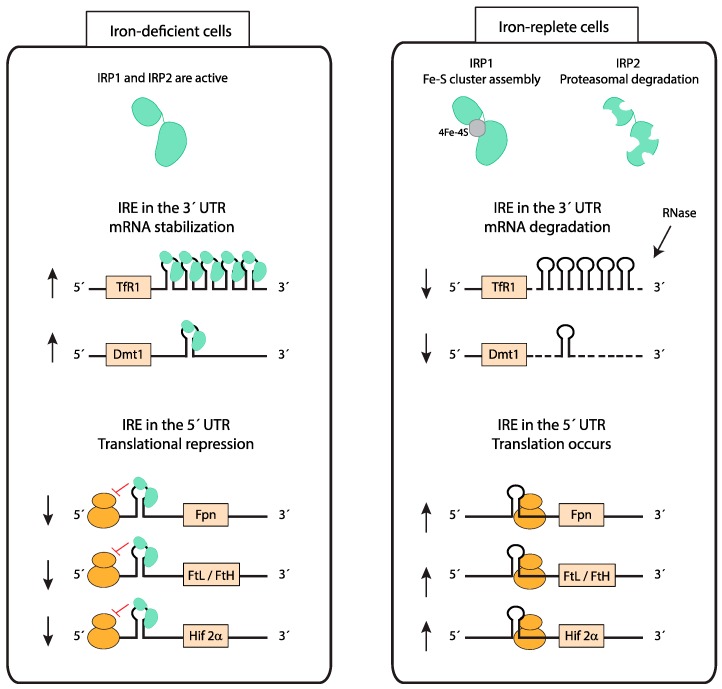
Cellular iron homeostasis: iron responsive element (IRE)/ iron regulatory protein (IRP) system. IRP1 and IRP2 bind to IREs present in either the 5’ untranslated regions (UTR) or 3’ UTR of mRNAs and regulate their translation and stability, respectively. In iron-depleted cells, IRPs bind to an IRE localized in the 5’ UTR of mRNAs to repress translation, while IRP binding to IREs in the 3’ UTR stabilizes mRNAs. In iron-replete cells, IRP1 switches from its IRE-binding form to a Fe-S cluster containing aconitase and IRP2 is degraded. The lack of IRP binding to IREs allows for the translation of mRNAs containing an IRE in the 5’ UTR and degradation of mRNAs containing IREs in the 3’ UTR. This mechanism counterbalances both cellular iron deficiency and iron overload. (Fpn—ferroportin; FtL—ferritin light chain; FtH—ferritin heavy chain; HIF-2α—hypoxia-inducible factor-2α).

**Table 1 pharmaceuticals-12-00005-t001:** Lung diseases and other diseases with a pathological lung phenotype associated with disturbed iron homeostasis.

Disease	Primary Lung Dysfunction	Systemic Iron Availability	Lung Iron Availability	References
Asthma	Obstructive	↔ to ↓	↓	[156,158,159]
ARDS	Shunt, V/Q mismatch	↔	↑	[9,19,156,157,160]
CF	Obstructive	↔ to ↓	↑	[19,156,161,162]
CMS	V/Q mismatch	↔ to ↓ ^1^	↓ ^1^	[133,135,163,164]
COPD	Obstructive	↔ to ↓	↑	[155,156,157,165,166]
HAPE	Diffusion Limitation	↔ to ↓	↓	[167,168]
IPF	Restrictive	↔	↑	[156,169,170]
Lung CA	N.A.	↔ to ↓	↑	[156,157,171,172,173,174,175,176]
PAP	Shunt	↔	↑	[19,156,177,178]
PH	V/Q mismatch	↔ to ↓	↓	[133,135,179,180,181,182,183,184,185]
TM	Restrictive	↔ to ↑ ^1^	↑ ^1^	[98,100,186,187,188,189]

ARDS: Acute respiratory distress syndrome; CF: Cystic fibrosis; CMS: Chronic mountain sickness; COPD: Chronic obstructive pulmonary disease; HAPE: High-altitude pulmonary edema; IPF: Idiopathic pulmonary fibrosis; Lung CA: Lung cancer; PAP: Pulmonary alveolar proteinosis; PH: Pulmonary hypertension; TM: Thalassemia Major; V/Q mismatch: Ventilation/Perfusion mismatch; N.A.: Not applicable; ↔ normal iron availability; ↓ reduced iron availability; ↑ increased iron availability; ^1^ Treatment induced.

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
