# Peer review of "Iron Homeostasis in the Lungs—A Balance between Health and Disease"

_pharmaceuticals, 2019, doi:10.3390/ph12010005_

Round 1

Reviewer 1 Report

This is a marvelously written comprehensive review of an important emerging area in lung physiology and pulmonary medicine by a group of highly expert authors in the fields of hematology, iron metabolism and hypoxia biology.

Major comments.

1. Figures 1 and 2 are out of order.

2. page 4.  The normal alveolar lining fluid is anywhere between 3-5 times more acidic than extracellular fluid.  Could you comment on what implications this might have on iron status in the lung?  Also the loss of a normal alkaline bronchial lining fluid in CF may be important. 

3. Page 11 and 12;  On line 436 you state that airway function must be regulated at high altitude.  There is little to no evidence that oxygen regulates airway tone, although CO2 does as it does also for the vasculature and parenchymal compliance.   It is also not clear to me what you mean by 'quality of this process is reflected by a high ventilation perfusion match'.  Hypoxia-hypoxemia-mediated increased ventilation via the carotid bodies leads to the high V/Q ratio of the lungs. 

4. page 13.  You really do not discuss AMS at all in relation to iron. 

There is one paper in the literature in 2015 from a Chinese group that showed no effect of iron supplementation on AMS, while in 2011 that Robbins group reported it did. 

5. page 13.   HAPE is not really a severe manifestation of AMS.  The one is a pulmonary hemodynamic problem and the other a cerebral intolerance of hypoxia.  (Swenson and Bartsch, Comp Physiol).  Any linkage is probably only that many people get AMS so that many people with HAPE get AMS.  However HAPE can occur without AMS. 

6. page 13.  CMS is not a chronic form of AMS. 

7. Table 1 needs the following changes.  First AB (also on page 15) is not a well recognized term for Asthma. I suggest you just use the term asthma and delete this abbreviation.  LUCA for lung cancer is also not widely used and I would use Lung CA instead.  The primary  dysfunction in ARDS is shunt and low ventilation/perfusion problems, I don't think there is any evidence for diffusion limitation in CMS, hypoxemia may more be related to relative hypoventilation brought on by polycythemia and V/Q mismatch. PAP is mostly a shunt problem and PH a V/Q mismatch problem or an intracardiac shunt if a PFO exists. 

8. page 14, line 561-4.  Do we know if CMS patients beyond the extra iron in their polycythymic blood have inappropriate iron stores?  This would be key issue in consideration of using agents that affect iron metabolism. From my reading of the Frise paper, the ferritin concentrations in the CMS group did not look all that high.

9. page 16. line 652. You might want to bring in the study of Goss et al 2018 in Sci Trans Med in using gallium to disrupt iron homeostasis in Pseudomonas to reduce bacterial growth and biofilm production.

Minor comments

1. The author's command of English is excellent, but there are a few places where the grammar and word choice might be improved.

- everywhere that a numerical value is given, there should be a space between the value and units.  e.g page 1, line 41; 20-25 mg

2. page 2, line 60-62.  Isn't recycling of senescent rbcs only able to maintain the pool of iron needed for replacement new Hb and rbc formation?  I don't see how recovery of this iron can be used for other purposes.  

3. page 2, line 68; shouldn't endocytosis come first before ubiquitination?

4. page 3, line 101; delete the unnecessary use of 'an'

5. page 5, line 158.  I believe 'gram' should be capitalized to recognize the man who discovered this classification of bacteria.

6. page, line 183.   Although you later in the paragraph define these abbreviations, it is best to do so the at their first usage. 

7. page 7, line 244. add 'with' before 'anemia' and correct spelling of 'somel to 'some'

8. page 8, line 308 and page 10, line 385.  There is an unnecessary tendency to make verbs out of nouns in English when sufficient words or wording already exists.  Using impact as a verb is such an example.  Either use 'affect' or write 'have an impact on'. 

9. page 10, line 389. change 'proof' to 'prove'

10. page 10. line 392.  the use of a.s.l. is unneccesary. 

11. page 10, line 400; change 'map-adaptions' to 'maladaptations' and elsewhere.

12. page 10, line 414. Does sweating really cause a significant loss of iron.  

13. page 11, line 451.  change 'nominated them' to 'led to their nomination' 

14. page 13, line 'of' to ' 'with' and add 'at rest' after 'dyspnea'

15. page 14. line 550. change to H > 210 g/l.  The capitalization of L for liter is unnecessary, since we do not do the same for other common units for mass, time or length.  The argument that L might be mistaken for the number 1 is weak, since in the context it would make no sense. 

16. page 14, line 570.  change 'unbalance' to 'inbalance' 

17. page 14, line 588.  Change 'ameliorated' to 'reduced' 

18. page 15, line 601.  Delete 'The' before 'COPD' - it is unnecessary. 

19. page 15, line 607, change 'just' to 'only'

20. page 15, lines 609-11.  In what direction do they change?

21. page 15, line 611-613  Is this due to hepicidin elevation? 

22.  page 15, lines 630-636.  Isn't the inflammatory state of asthma with anemia also very likely due to up regulation of hepcidin? 

23. page 16, line 672.  change 'to mention' to 'mentioning'

24. page 16, line 693.  I would add also low grade bleeding as a source of iron.

25. page 17,  add 'with' before 'higher'

26. page 17, line 700, change 'for' to 'to'

27. page 17, line 711.  Given the complexity and lack of sufficient data, it might be better to add 'careful' before 'use'

28. page 17, line 731. I would add 'no or only' before 'a'

29. page 17, line 742-  use maladaptation instead of mal-adaption.

30. page 17, line 744, suggest changing 'interference' to 'suppression and/or enhancement'.

31. page 18.  this last sentence is quite long. I suggest that 'as promising therapeutic potential for the future' is unnecessary.

Author Response

Response to Reviewer 1 Comments

Comments and Suggestions for Authors

Reviewer 1:

This is a marvelously written comprehensive review of an important emerging area in lung physiology and pulmonary medicine by a group of highly expert authors in the fields of hematology, iron metabolism and hypoxia biology.

Response: We thank the reviewer for her/his positive comments

Major comments.

1. Figures 1 and 2 are out of order.

Response 1: We thank the reviewer for the correction. We have re-arranged the figures in the revised version of the paper accordingly.

2. page 4.  The normal alveolar lining fluid is anywhere between 3-5 times more acidic than extracellular fluid.  Could you comment on what implications this might have on iron status in the lung?  Also the loss of a normal alkaline bronchial lining fluid in CF may be important. 

Response 2: This is an interesting point noted by the referee. Indeed, the pH of lung fluids (e.g. the alveolar lining fluid, ALF) may affect the complex process of spontaneous oxidation of ferrous iron (Fe (II)) to ferric iron (Fe (III)) following a sigmoid shaped oxidation rate in relation to pH (Morgan and Lahav 2007). Therefore, changes in pH may alter the concentration/availability of the two iron redox-states within the ALF, which may subsequently affect trans-epithelial iron uptake and intracellular iron availability within the lung. The presence of ALF acidification in CF patients may limit Fe (II) oxidation, which may lead to increased availability of Fe (II) that potentially increases pathogen growth, e.g. of P. aeruginosa (Hunter et al. 2013). However, we would like to mention that the heterogeneity of study results in measuring the ALF pH in CF patients (e.g. in children with CF), potentially due to the technical challenge may suggest to interpret mechanistic implications with caution (Schultz et al. 2017).

We have included part of this explanation into the Ms (end of section 2.3.3).

3. Page 11 and 12;  On line 436 you state that airway function must be regulated at high altitude.  There is little to no evidence that oxygen regulates airway tone, although CO2 does as it does also for the vasculature and parenchymal compliance.   It is also not clear to me what you mean by 'quality of this process is reflected by a high ventilation perfusion match'.  Hypoxia-hypoxemia-mediated increased ventilation via the carotid bodies leads to the high V/Q ratio of the lungs. 

Response 3: We agree with the reviewer’s statement that CO2 is an important regulator of airway tone and also for the compliance of the pulmonary vasculature and parenchyma We would like to mention here that there is evidence from animal (Wetzel et al. 1992; Cardell et al. 1998) and human studies that acute exposure to hypoxia leads to bronchodilation even when CO2 levels are kept constant (isocapnic) (Julià-Serdà et al. 1993). Furthermore, exposure to prolonged hypoxia leads to airway proliferation in an HIF-dependent manner (Urrutia and Aragonés 2018), which can be considered as an ‘adaptive’ airway response to hypoxia. Indeed, high altitude residents seem to have an increased airway capacity (as reflected by increased total lung capacity) in comparison to lowlanders (Havryk, Gilbert, and Burgess 2002; Droma et al. 1991). In agreement with the reviewer’s statement the carotid bodies (CBs) are the main vascular oxygen sensors and essential for the hypoxic ventilatory response (HVR) (López-Barneo et al. 2016), however the neuroepithelial bodies (NEBs) within the lungs are also considered as multimodal chemo-sensors being able to sense both O2 (Youngson et al. 1993) and CO2(H+) (Livermore et al. 2015) and as the CBs do (López-Barneo et al. 2016) the NEBs proliferate and grow as an adaptive response to prolonged hypoxic exposure (Pan et al. 2016).

We have rephrased the following sentence questioned by the reviewer: 'quality of this process is reflected by a high ventilation perfusion match'. In the revised version it now reads ‘the functional quality of this process is reflected by an improved ventilation/perfusion (V/Q) balance’.

4. page 13.  You really do not discuss AMS at all in relation to iron. 

There is one paper in the literature in 2015 from a Chinese group that showed no effect of iron supplementation on AMS, while in 2011 that Robbins group reported it did. 

Response 4: The reviewer is correct. As also stated by him/her in comment 5, AMS can be seen as a ‘cerebral intolerance to hypoxia’ and we therefore did not discuss it in further details as the primary focus of our review is on lung related diseases and the potential role of disturbed iron homeostasis (for further details see also answer to comment 5.).

5. page 13.   HAPE is not really a severe manifestation of AMS.  The one is a pulmonary hemodynamic problem and the other a cerebral intolerance of hypoxia.  (Swenson and Bartsch, Comp Physiol).  Any linkage is probably only that many people get AMS so that many people with HAPE get AMS.  However HAPE can occur without AMS. 

Response 5: We apologize for the confusion we caused. Indeed, HAPE can even occur without symptoms/signs of AMS. We therefore changed the text section accordingly and added the suggested literature. The text section now reads:

 ‘5.1. Acute high altitude illnesses and high altitude pulmonary edema (HAPE)

The classical high altitude illnesses are also comprised under the term ‘mountain sicknesses’. Acute mountain sickness (AMS) is a common manifestation of high altitude illness and related to a cerebral intolerance to hypoxia, which is also the case in high altitude cerebral edema (HACE), a severe and life threating condition [166,189]. In the lungs, the development of high altitude pulmonary edema (HAPE) can occur in unacclimatized healthy individuals at high altitude (2500 m above sea level) within 1-5 days upon arrival [166,189,190]’.

6. page 13.  CMS is not a chronic form of AMS. 

Response 6: We thank the reviewer for pointing to this error. We’ve changed the text accordingly (see also answer 5).

7. Table 1 needs the following changes.  First AB (also on page 15) is not a well recognized term for Asthma. I suggest you just use the term asthma and delete this abbreviation.  LUCA for lung cancer is also not widely used and I would use Lung CA instead.  The primary  dysfunction in ARDS is shunt and low ventilation/perfusion problems, I don't think there is any evidence for diffusion limitation in CMS, hypoxemia may more be related to relative hypoventilation brought on by polycythemia and V/Q mismatch. PAP is mostly a shunt problem and PH a V/Q mismatch problem or an intracardiac shunt if a PFO exists. 

Response 7: We thank the reviewer for the valuable corrective suggestions concerning the Table1. We’ve adjusted the table accordingly (see Table1 of revised paper).

8. page 14, line 561-4.  Do we know if CMS patients beyond the extra iron in their polycythymic blood have inappropriate iron stores?  This would be key issue in consideration of using agents that affect iron metabolism. From my reading of the Frise paper, the ferritin concentrations in the CMS group did not look all that high.

Response 8: We thank the reviewer for the valuable contribution. Indeed, the reported ferritin levels of CMS patients prior to the iron infusion in the study performed by Smith et al. (Smith et al. 2009) to which the Frise paper (Frise and Robbins 2015) also refers, are rather ‘low than high’. The disturbance in iron stores in CMS patients is primarily related to clinical/therapeutic interventions (Villafuerte and Corante 2016), see also Table1, footnote 1)) to lower the hematocrit/hemoglobin levels in these patients using iron chelator therapy and phlebotomy to counterbalance the excessive erythrocytosis. In the study of Smith et al. (Smith et al. 2009) repeated venesections (days 1-4, total blood volume loss: 2L) were performed prior to the iron/Placebo infusion. Thus ‘low’ ferritin levels are expected in these treated CMS patients as an indication of relative iron deficiency induced by blood volume loss in the presence of increased erythropoietic activity. The presence of iron deficiency may lead to worsening of PH in CMS patients as suggested by the study results of Smith et al. (Smith et al. 2009). As further stated by the same study authors a monitoring of iron stores may therefore be clinically helpful to improve disease related conditions (e.g. PH) in CMS patients.

9. page 16. line 652. You might want to bring in the study of Goss et al 2018 in Sci Trans Med in using gallium to disrupt iron homeostasis in Pseudomonas to reduce bacterial growth and biofilm production.

Response 9: We thank the reviewer for the interesting literature suggestion. We therefore added the following small section under 5.6 Cystic Fibrosis addressing this paper:

‘Alternatively, a recent study showed that it is also possible to directly target bacterial iron homeostasis to control airway P. aeruginosa infections via the administration of the metal gallium [210]. Gallium is taken up by bacteria instead of iron because bacterial uptake systems cannot differentiate them. Gallium is then incorporated into iron-containing proteins but, since it cannot be reduced under physiological conditions, it impairs their normal activity and therefore bacterial survival and proliferation.’

Minor comments

1. The author's command of English is excellent, but there are a few places where the grammar and word choice might be improved.

- everywhere that a numerical value is given, there should be a space between the value and units.  e.g page 1, line 41; 20-25 mg

Response 1: The space was added.

2. page 2, line 60-62.  Isn't recycling of senescent rbcs only able to maintain the pool of iron needed for replacement new Hb and rbc formation?  I don't see how recovery of this iron can be used for other purposes.  

Response 2: Iron recycled by macrophages goes back into circulation and therefore is available to any cell in the body. It is true that the production of new RBCs in the bone marrow is the major site for iron consumption in the body but, even though at a much lower level, other cells also need iron (e.g. for ribonucleotide reductase which is essential for DNA synthesis and cytochromes in the mitochondria).

3. page 2, line 68; shouldn't endocytosis come first before ubiquitination?

Response 3: Upon hepcidin binding, the ubiquitination of multiple lysines within the cytoplasmatic loop of ferroportin is necessary to induce its endocytosis (Qiao et al. 2012).

4. page 3, line 101; delete the unnecessary use of 'an'

Response 4: It was deleted.

5. page 5, line 158.  I believe 'gram' should be capitalized to recognize the man who discovered this classification of bacteria.

Response 5: It was corrected.

6. page, line 183.   Although you later in the paragraph define these abbreviations, it is best to do so the at their first usage. 

Response 6: The abbreviations are now defined at their first usage.

7. page 7, line 244. add 'with' before 'anemia' and correct spelling of 'somel to 'some'

Response7: It was added and corrected.

8. page 8, line 308 and page 10, line 385.  There is an unnecessary tendency to make verbs out of nouns in English when sufficient words or wording already exists.  Using impact as a verb is such an example.  Either use 'affect' or write 'have an impact on'. 

Response 8: It was corrected.

9. page 10, line 389. change 'proof' to 'prove'

Response 9: It was changed.

10. page 10. line 392.  the use of a.s.l. is unneccesary. 

Response 10: This abbreviation was removed and replaced when needed by “above sea level”

11. page 10, line 400; change 'map-adaptions' to 'maladaptations' and elsewhere.

Response 11: It was changed.

12. page 10, line 414. Does sweating really cause a significant loss of iron.  

Response 12: The loss of iron through sweating during exercise may indeed be considered a ‘minor’ among several (as mentioned in the text) contributing factors in this context. Generally, the loss of iron through sweating is dependent on the intensity of physical activity and also on the environmental conditions such as the ambient temperature (Waller and Haymes 1996). However, in subjects with a priori low iron stores (e.g. female athletes with low dietary iron intake) who are exposed to frequent prolonged exercise bouts, the iron loss through sweating may become the ‘tipping point’ (Lamanca et al. 1988).

13. page 11, line 451.  change 'nominated them' to 'led to their nomination' 

Response 13: It was changed.

14. page 13, line 'of' to ' 'with' and add 'at rest' after 'dyspnea'

Response 14: It was changed.

15. page 14. line 550. change to H > 210 g/l.  The capitalization of L for liter is unnecessary, since we do not do the same for other common units for mass, time or length.  The argument that L might be mistaken for the number 1 is weak, since in the context it would make no sense. 

Response 15: It was corrected.

16. page 14, line 570.  change 'unbalance' to 'inbalance' 

Response 16: It was changed.

17. page 14, line 588.  Change 'ameliorated' to 'reduced' 

Response 17: It was changed.

18. page 15, line 601.  Delete 'The' before 'COPD' - it is unnecessary. 

Response 18: It was deleted.

19. page 15, line 607, change 'just' to 'only'

Response 19: It was changed.

20. page 15, lines 609-11.  In what direction do they change?

Response 20: We’ve changed the text accordingly. It now reads ‘A recent large cohort study found that in both smokers and non-smokers certain markers of iron homeostasis such as serum ferritin concentration, serum iron concentration and transferrin saturation were associated with critical parameters of lung function (e.g. positive correlation for all three markers in both groups combined with forced vital capacity and forced expiratory capacity) [205].’

21. page 15, line 611-613  Is this due to hepcidin elevation?

Response 21: Yes, the anemia observed in COPD patients is likely the result of higher levels of hepcidin. We mention this in the next paragraph.

We have now added some text to make it clear. It now reads: “It has been recently proposed that in patients with COPD pro-inflammatory cytokines (e.g. IL-6) trigger increased hepatic hepcidin expression and secretion (Cloonan et al. 2017). This causes FPN degradation and subsequently reduces cellular iron export into the blood stream, likely resulting in the systemic iron deficiency and anemia observed in these patients

22.  page 15, lines 630-636.  Isn't the inflammatory state of asthma with anemia also very likely due to up regulation of hepcidin?

Response 22: Correct, inflammation triggered up-regulation of hepcidin might be an important factor contributing to the setting of absolute or functional iron deficiency in patients with asthma. The review of Ghio AJ et al. (Ghio 2016), which we’d referred to in this section primarily addressed the role of hepcidin in the ‘acute phase response’ due to acute infections leading to decreased iron availability for the lung pathogen. Future studies are warranted to specifically address this aspect.

23. page 16, line 672.  change 'to mention' to 'mentioning'

Response 23: It is changed.

24. page 16, line 693.  I would add also low grade bleeding as a source of iron.

Response 24: We’ve added this aspect in the revised version of the review. It now reads ‘A potential source of iron in ARDS may derive from ‘low grade bleeding’ and hemorrhage, which might be present in certain subsets of this heterogenous pathological lung condition [214].’

25. page 17,  add 'with' before 'higher'

Response 25: It is added.

26. page 17, line 700, change 'for' to 'to'

Response 26: It was changed.

27. page 17, line 711.  Given the complexity and lack of sufficient data, it might be better to add 'careful' before 'use'

Response 27: It was added.

28. page 17, line 731. I would add 'no or only' before 'a'

Response 28: It was added.

29. page 17, line 742-  use maladaptation instead of mal-adaption.

Response 29: It was corrected.

30. page 17, line 744, suggest changing 'interference' to 'suppression and/or enhancement'.

Response 30: It was replaced.

31. page 18.  this last sentence is quite long. I suggest that 'as promising therapeutic potential for the future' is unnecessary.

Response 31: We removed this part of the sentence.

Submission Date

20 November 2018

Date of this review

04 Dec 2018 02:14:00

 References:

Cardell, L. O., Y. P. Lou, K. Takeyama, I. F. Ueki, J. Lausier, and J. A. Nadel. 1998. “Carbon Monoxide, a Cyclic GMP-Related Messenger, Involved in Hypoxic Bronchodilation in Vivo.” Pulmonary Pharmacology & Therapeutics 11 (4): 309–15. https://doi.org/10.1006/pupt.1998.0152.

Cloonan, Suzanne M., Sharon Mumby, Ian M. Adcock, Augustine M. K. Choi, Kian Fan Chung, and Gregory J. Quinlan. 2017. “The ‘Iron’-y of Iron Overload and Iron Deficiency in Chronic Obstructive Pulmonary Disease.” American Journal of Respiratory and Critical Care Medicine 196 (9): 1103–12. https://doi.org/10.1164/rccm.201702-0311PP.

Droma, T., R. G. McCullough, R. E. McCullough, J. G. Zhuang, A. Cymerman, S. F. Sun, J. R. Sutton, and L. G. Moore. 1991. “Increased Vital and Total Lung Capacities in Tibetan Compared to Han Residents of Lhasa (3,658 M).” American Journal of Physical Anthropology 86 (3): 341–51. https://doi.org/10.1002/ajpa.1330860303.

Frise, Matthew C., and Peter A. Robbins. 2015. “The Pulmonary Vasculature--Lessons from Tibetans and from Rare Diseases of Oxygen Sensing.” Experimental Physiology 100 (11): 1233–41. https://doi.org/10.1113/expphysiol.2014.080507.

Ghio, Andrew J. 2016. “Asthma as a Disruption in Iron Homeostasis.” Biometals: An International Journal on the Role of Metal Ions in Biology, Biochemistry, and Medicine 29 (5): 751–79. https://doi.org/10.1007/s10534-016-9948-y.

Havryk, Adrian P., Mark Gilbert, and Keith R. Burgess. 2002. “Spirometry Values in Himalayan High Altitude Residents (Sherpas).” Respiratory Physiology & Neurobiology 132 (2): 223–32.

Hunter, Ryan C., Fadi Asfour, Jozef Dingemans, Brenda L. Osuna, Tahoura Samad, Anne Malfroot, Pierre Cornelis, and Dianne K. Newman. 2013. “Ferrous Iron Is a Significant Component of Bioavailable Iron in Cystic Fibrosis Airways.” MBio 4 (4). https://doi.org/10.1128/mBio.00557-13.

Julià-Serdà, G., N. A. Molfino, H. G. Furlott, P. A. McClean, A. S. Rebuck, V. Hoffstein, A. S. Slutsky, N. Zamel, and K. R. Chapman. 1993. “Tracheobronchial Dilation during Isocapnic Hypoxia in Conscious Humans.” Journal of Applied Physiology (Bethesda, Md.: 1985) 75 (4): 1728–33. https://doi.org/10.1152/jappl.1993.75.4.1728.

Lamanca, J. J., E. M. Haymes, J. A. Daly, R. J. Moffatt, and M. F. Waller. 1988. “Sweat Iron Loss of Male and Female Runners during Exercise.” International Journal of Sports Medicine 9 (1): 52–55. https://doi.org/10.1055/s-2007-1024978.

Livermore, S., Y. Zhou, J. Pan, H. Yeger, C. A. Nurse, and E. Cutz. 2015. “Pulmonary Neuroepithelial Bodies Are Polymodal Airway Sensors: Evidence for CO2/H+ Sensing.” American Journal of Physiology. Lung Cellular and Molecular Physiology 308 (8): L807-815. https://doi.org/10.1152/ajplung.00208.2014.

López-Barneo, José, David Macías, Aida Platero-Luengo, Patricia Ortega-Sáenz, and Ricardo Pardal. 2016. “Carotid Body Oxygen Sensing and Adaptation to Hypoxia.” Pflugers Archiv: European Journal of Physiology 468 (1): 59–70. https://doi.org/10.1007/s00424-015-1734-0.

Morgan, Barak, and Ori Lahav. 2007. “The Effect of PH on the Kinetics of Spontaneous Fe(II) Oxidation by O2 in Aqueous Solution--Basic Principles and a Simple Heuristic Description.” Chemosphere 68 (11): 2080–84. https://doi.org/10.1016/j.chemosphere.2007.02.015.

Pan, Jie, Tammie Bishop, Peter J. Ratcliffe, Herman Yeger, and Ernest Cutz. 2016. “Hyperplasia and Hypertrophy of Pulmonary Neuroepithelial Bodies, Presumed Airway Hypoxia Sensors, in Hypoxia-Inducible Factor Prolyl Hydroxylase-Deficient Mice.” Hypoxia (Auckland, N.Z.) 4: 69–80. https://doi.org/10.2147/HP.S103957.

Qiao, Bo, Priscilla Sugianto, Eileen Fung, Alejandro Del-Castillo-Rueda, Maria-Josefa Moran-Jimenez, Tomas Ganz, and Elizabeta Nemeth. 2012. “Hepcidin-Induced Endocytosis of Ferroportin Is Dependent on Ferroportin Ubiquitination.” Cell Metabolism 15 (6): 918–24. https://doi.org/10.1016/jNaNet.2012.03.018.

Schultz, André, Ramaa Puvvadi, Sergey M. Borisov, Nicole C. Shaw, Ingo Klimant, Luke J. Berry, Samuel T. Montgomery, et al. 2017. “Airway Surface Liquid PH Is Not Acidic in Children with Cystic Fibrosis.” Nature Communications 8 (1): 1409. https://doi.org/10.1038/s41467-017-00532-5.

Smith, Thomas G., Nick P. Talbot, Catherine Privat, Maria Rivera-Ch, Annabel H. Nickol, Peter J. Ratcliffe, Keith L. Dorrington, Fabiola León-Velarde, and Peter A. Robbins. 2009. “Effects of Iron Supplementation and Depletion on Hypoxic Pulmonary Hypertension: Two Randomized Controlled Trials.” JAMA 302 (13): 1444–50. https://doi.org/10.1001/jama.2009.1404.

Urrutia, Andrés A., and Julián Aragonés. 2018. “HIF Oxygen Sensing Pathways in Lung Biology.” Biomedicines 6 (2). https://doi.org/10.3390/biomedicines6020068.

Villafuerte, Francisco C., and Noemí Corante. 2016. “Chronic Mountain Sickness: Clinical Aspects, Etiology, Management, and Treatment.” High Altitude Medicine & Biology 17 (2): 61–69. https://doi.org/10.1089/ham.2016.0031.

Waller, M. F., and E. M. Haymes. 1996. “The Effects of Heat and Exercise on Sweat Iron Loss.” Medicine and Science in Sports and Exercise 28 (2): 197–203.

Wetzel, R. C., C. J. Herold, E. A. Zerhouni, and J. L. Robotham. 1992. “Hypoxic Bronchodilation.” Journal of Applied Physiology (Bethesda, Md.: 1985) 73 (3): 1202–6. https://doi.org/10.1152/jappl.1992.73.3.1202.

Youngson, C., C. Nurse, H. Yeger, and E. Cutz. 1993. “Oxygen Sensing in Airway Chemoreceptors.” Nature 365 (6442): 153–55. https://doi.org/10.1038/365153a0.

Reviewer 2 Report

The review is clearly written and comprehensively discussed the regulation of lung iron homeostasis and its deregulation in lung diseases. The topic is extremely interesting and of potential interest for a broad audience.

Minor comments:

(a)    Figure 1 and 2 are inverted. The first picture refers to “Lung iron homeostasis” instead of “Systemic Iron Homeostasis” as indicated in the text (lane 66, lane 115, …) and figure legends.

(b)    Please, specify blood vessels in the second picture, like you have done in Figure 1.

(c)     Paragraph 2. Lung Iron Homeostasis

Another source of iron is represented by heme released following intravascular hemolysis. Are heme transporters and/or heme oxygenase expressed in airway epithelial cells and/or alveolar macrophages?

The authors described pulmonary  dysfunction in patients with thalassemia major (TM). Are there any additional data indicating increased susceptibility to lung diseases in patients with other hemolytic diseases?

Author Response

Response to Reviewer 2 Comments

Comments and Suggestions for Authors (Reviewer 2)

The review is clearly written and comprehensively discussed the regulation of lung iron homeostasis and its deregulation in lung diseases. The topic is extremely interesting and of potential interest for a broad audience.

Response: We thank the reviewer for her/his positive comments.

Minor comments:

(a)    Figure 1 and 2 are inverted. The first picture refers to “Lung iron homeostasis” instead of “Systemic Iron Homeostasis” as indicated in the text (lane 66, lane 115, …) and figure legends.

Response (a): We thank the reviewer for the correction. We have re-arranged the figures in the revised version of the paper accordingly.

(b)    Please, specify blood vessels in the second picture, like you have done in Figure 1.

Response (b): We thank the reviewer for the corrective suggestion. We’ve modified Figure 2 accordingly.

(c)     Paragraph 2. Lung Iron Homeostasis

Another source of iron is represented by heme released following intravascular hemolysis. Are heme transporters and/or heme oxygenase expressed in airway epithelial cells and/or alveolar macrophages?

Response: We have now included a paragraph related to lung exposure to heme in the section 2.3.1. Control of pulmonary iron uptake.

The authors described pulmonary dysfunction in patients with thalassemia major (TM). Are there any additional data indicating increased susceptibility to lung diseases in patients with other hemolytic diseases?

Response: We would like to thank the reviewer for this suggestion. There is indeed evidence for increased susceptibility to lung diseases in patients with sickle cell disease. We have added this to the section 5.8 Other Diseases.

Submission Date

20 November 2018

Date of this review

10 Dec 2018 15:55:43